# Proposal of *Brotulella* n. gen. for Monogeneans from the Gills of the Pacific Bearded Brotula *Brotula clarkae* Hubbs, 1944 (*Ophidiiformes: Ophidiidae*) Based on Morphological and Molecular Evidence

Celso Luis Cruces [1,2], Raquel Simões [3], Arnaldo Maldonado Júnior [4], Ruperto Severino [2], Jhon Darly Chero [2] and José Luis Luque [3,*]

1   Graduate Program in Animal Biology, Federal Rural University of Rio de Janeiro, Seropédica 23890-000, RJ, Brazil
2   Invertebrate Zoology Laboratory, Academic Department of Zoology, Faculty of Biological Sciences, National University of San Marcos, Lima 15081, Peru
3   Department of Animal Parasitology, Federal Rural University of Rio de Janeiro, Seropédica 23890-000, RJ, Brazil
4   Laboratory of Biology and Parasitology of Wild Mammalian Reservoirs, Oswaldo Cruz Institute, Rio de Janeiro 22061-040, RJ, Brazil
*   Correspondence: luqueufrrj@gmail.com

**Abstract:** Based on morphological and molecular data, *Brotulella* n. gen. is proposed to accommodate the dactylogyrid monogeneans *Brotulella laurafernandae* n. sp. (type species) and *Brotulella luisahelenae* n. sp. on the gill filaments of the Pacific bearded brotula *Brotula clarkae* Hubbs, 1944 (*Ophidiiformes: Ophidiidae*) from Puerto Pizarro in the Tumbes region (northern Peru). Species of the new genus are distinguished from all other dactylogyrids by the combination of the following features: (1) anchors with a stocking-shaped sheath associated with the distal end of superficial and deep roots, (2) tandem gonads, (3) a vas deferens looping left intestinal caecum, (4) a distally twisted male copulatory organ (MCO) with a delicate membranous accessory piece articulated to the shaft of the MCO, (5) a U-shaped ovary, (6) an almost sigmoid seminal vesicle, and (7) two prostatic reservoirs with thick muscular walls. A 28S ribosomal DNA-based phylogenetic analysis (Maximum likelihood and Bayesian inference) of sequences of two new species of *Brotulella* n. gen. from the Southeastern Pacific Ocean, along with sequences from closely related genera of the marine Dactylogyridae, supports the establishment of the new genus for the dactylogyrid parasites on the gills of the Pacific bearded brotula.

**Keywords:** *Brotulella* n. gen.; *Brotulella laurafernandae* n. sp.; *Brotulella luisahelenae* n. sp.; cusk-eels; gill ectoparasites; taxonomy; South America

**Key Contribution:** The manuscript introduces a new genus, *Brotulella* n. gen., to accommodate two newly discovered dactylogyrid monogeneans, *Brotulella laurafernandae* n. sp. and *Brotulella luisahelenae* n. sp., found on the gill filaments of the Pacific bearded brotula, *Brotula clarkae*. These parasites are distinguished by unique morphological features. Additionally, a phylogenetic analysis based on 28S ribosomal DNA sequences supports the establishment of this new genus, shedding light on the biodiversity of these marine parasites in the Southeastern Pacific Ocean.

## 1. Introduction

Peru, as other Neotropical countries, harbors a rich fauna of fishes, especially a large number of bony fishes [1]. Regarding its marine ichthyology diversity, a total of 698 fish species have been reported belonging to 388 genera and 138 families [2], most of them recorded from the northern sea, being one of the richest marine ecosystems in Peru [3].

Several parasitological surveys performed in northern Peru have found a rich fauna of the Dactylogyridea Bychowsky, 1937 from marine fishes, especially in serranid and sciaenid fishes [4,5].

Dactylogyridae Bychowsky, 1933, a diverse family of monogenean parasites, represents a significant component of parasitic fauna among fishes worldwide [6]. This group exhibits remarkable diversity in morphology, life cycles, and host specificity, making it imperative to comprehend their taxonomy and ecological roles [6]. Understanding the intricate relationships between Dactylogyridae species and their fish hosts is crucial for several reasons. Firstly, some of these parasites can induce pathogenic effects on their hosts, impacting fish health, growth, and even leading to mortality in some cases [7]. Secondly, their prevalence and abundance within aquatic ecosystems can serve as indicators of ecosystem health and environmental changes [8]. Moreover, due to their high host specificity and site selection within fish gills or skin, Dactylogyridae species can offer insights into fish migration patterns and habitat preferences [8].

The Pacific bearded brotula *Brotula clarkae* Hubbs, 1944 (*Ophidiiformes: Ophidiidae*) is a benthopelagic fish endemic to the Eastern Tropical Pacific [9–11], and is found deep in mud and in broken shell bottoms [12]. This species is distributed from the Gulf of California (USA) to Paita (Peru) [11,13,14]. Although numerous studies have focused on the reproductive biology of the Pacific bearded brotula [10,15,16], the parasitic fauna associated with *Br. clarkae* remains unknown to date. During a study on gill ectoparasites of marine fishes from Peru, two species belonging to the Dactylogyridae Bychowsky, 1937 were found on the Pacific bearded brotula. In this study, both species are examined based on morphological and molecular data.

## 2. Materials and Methods

### 2.1. Specimen Collection and Morphological Analyses

Two specimens of *Br. clarkae* were obtained from local fishermen from the Puerto Pizarro resort (3°29′ S, 80°24′ W), Tumbes, Peru, during a field expedition in February 2019. The hosts were dissected immediately, and the gill arches were removed and placed in vials containing sea water (60 °C). Each vial was vigorously shaken, and formalin was added to obtain a 4% solution. Some monogeneans were fixed directly in 70% ethanol and subsequently preserved in 90% ethanol until use. The anterior and posterior parts of these specimens were cut and used for morphological identification, whereas the middle parts were used in molecular procedures. In the laboratory, the contents of each vial were examined under a dissecting microscope and monogeneans were removed from the gills or sediment using small probes. Some specimens were stained with Gomori's trichrome, clarified in clove oil, and mounted in Canada balsam. Other specimens were mounted in Gray & Wess medium [17] for the study of sclerotised structures. Specimens were examined and photographed using a compound OlympusTM BX51 photomicroscope equipped with normal light and differential interference contrast microscopy (DIC) optics (Olympus Corporation, Tokyo, Japan). Drawings were made with the aid of a drawing tube. Measurements are in micrometers (μm), unless otherwise indicated, using straight-line distances between extreme points of the structures measured and are expressed as the range followed by the mean and number (n) of structures measured in parentheses. Body length represents the length of the body proper with the haptor. Numbering of haptoral-hook pairs followed the system of Mizelle [18] and Mizelle & Price [19]. Fishes were identified according to Chirichigno & Vélez [20]. The abbreviation *B.* for the parasite (*Brotulella* n. gen.) and *Br.* for the host (*Brotulla*) are used to avoid doubt as to the genera. The type-material was deposited in the Helminthological Collection in the Museum of Natural History at the San Marcos University (MUSM-HEL), Lima, Peru.

### 2.2. Molecular Characterization and Phylogenetic Analyses

Genomic DNA were isolated from one specimen of each new species of parasite using the Qiagen QIAamp DNA Mini Kit (Qiagen, Hilden, Germany), according to the

manufacturer's protocol. Partial 28S rRNA gene was amplified by polymerase chain reaction (PCR) using the primers C1 (5′–ACCCG CTGAA TTTAA GCAT–3′) and D2 (5′–TGGTC CGTGT TTCAA GAC–3′) [21–23]. The thermocycling profile, as described by Mendoza-Palmero et al. [24], consisted of an initial denaturation set at 95 °C for 3 min, followed by 34 cycles of amplification (94 °C for 30 s, 56 °C for 30 s and 72 °C for 1:30 min), with a final extension hold at 72 degrees C. for 4 min. Samples were sequenced using an ABI 3730 DNA analyzer of the RPT01A subunit for DNA sequencing of the Technological Platforms Network at the Oswaldo Cruz Foundation, Rio de Janeiro, Brazil. The resulting fragments were assembled into contiguous sequences in Geneious [25]. These sequences were deposited in GenBank SUB13983437 and SUB13983462. A total of 36 species, representing 18 genera of the Dactylogyridae infecting 27 marine fish host species, were included in the phylogenetic analyses. Additionally, 3 species of the Diplectanidae were utilized as an outgroup for comparison (see Table 1). Alignment and editing of sequences were performed using Clustal W in MEGA version 7.0 [26]. The Mega 7.0 software was used for construction of genetic distances using Kimura two parameters [27]. The online PhyML 3.0 software [28] was used to reconstruct phylogenies based on the maximum likelihood (ML) approach. The model of nucleotide evolution, general time reversible (GTR), was selected with Smart Model Selection (SMS), run in PhyML [29], using the Bayesian information criterion. Node support was computed by the approximate likelihood-ratio test for branches (aLRT) [30] and by nonparametric bootstrap percentages (ML-BP) with 1000 pseudoreplicates. Bayesian phylogenetic inference (BI) was carried out using MrBayes version 3.2.6 [31] in XSEDE using the CIPRES Science Gateway [32], with the GTR + G model. Markov chain Monte Carlo samplings for each matrix were performed for 10,000,000 generations with four simultaneous chains in two runs. Branch supports in Bayesian trees by Bayesian posterior probabilities (BPP) were assessed from trees that were sampled every 100 generations, after removal of a burn-in fraction of 25%. The sequences from GenBank that were included for the phylogenetic analyses are listed in Table 1 as well as sequences for out-groups.

**Table 1.** List of dactylogyrid species included in the phylogenetic analyses. Sequences obtained for the present study are in bold.

| Species | Host Species | Locality | GenBank ID |
|---|---|---|---|
| Dactylogyridae | | | |
| *Bravohollisia roseta* | *Pomadasys argenteus* | Zhanjiang, Guangdong, China | KJ571011 |
| *Bravohollisia tecta* | *Pomadasys maculatus* | Zhanjiang, Guangdong, China | KJ571012 |
| **Brotulella laurafernandae n. sp.** | **Brotula clarkae** | **Off Puerto Pizarro resort, Peru** | OR860318 |
| **Brotulella luisahelenae n. sp.** | **Brotula clarkae** | **Off Puerto Pizarro resort, Peru** | OR860321 |
| *Caballeria intermedius* | *Pomadasys argenteus* | Zhanjiang, Guangdong, China | KJ571013 |
| *Chauhanellus auriculatum* | *Arius maculatus* | Malaysia | MN108169 |
| *Chauhanellus boegeri* | *Genidens genidens* | Baia de Antonia, Brazil | KP056241 |
| *Euryhaliotrema longibaculum* | *Lutjanus synagris* | Campeche Bank, Mexico | MG586863 |
| *Euryhaliotrema carbuncularium* | *Archosargus rhomboidalis* | Campeche Bank, Mexico | MG586875 |
| *Euryhaliotrema tubocirrus* | *Lutjanus griseus* | Campeche Bank, Mexico | MG586848 |
| *Haliotrema bilobatum* | *Drepane punctata* | Malaysia | MG593837 |
| *Haliotrema cromileptis* | *Epinephelus coioides* | Nha Trang Bay, Vietnam | EU523146 |
| *Haliotrema johnstoni* | *Upeneus luzonius* | Haikou, Hainan Province, China | DQ157664 |
| *Haliotrematoides spinatus* | *Lutjanus guttatus* | Pacific Ocean, off Mexico | HQ615995 |
| *Haliotrematoides heteracantha* | *Lutjanus synagris* | Campeche Bank, Mexico | MG586855 |
| *Haliotrematoides magnigastrohamus* | *Lutjanus synagris* | Campeche Bank, Mexico | MG586849 |
| *Lethrinitrema fleti* | *Lethrinus nebulosus* | China | EU836203 |
| *Lethrinitrema grossecurvitubus* | *Lethrinus nebulosus* | China | EU836204 |
| *Ligophorus uruguayensis* | *Mugil liza* | Brazil | KF442630 |
| *Ligophorus kaohsianghsieni* | *Planiliza haematocheilus* | Russia | KY979159 |
| *Metahaliotrema scatophogi* | *Scatophagus argus* | Panyu, Guangdong, China | DQ157646 |
| *Metahaliotrema digyroides* | *Gerres macrosoma* | China | DQ537377 |

**Table 1.** *Cont.*

| Species | Host Species | Locality | GenBank ID |
| --- | --- | --- | --- |
| *Mexicana rubra* | *Orthopristis ruber* | Coast of Rio de Janeiro, Brazil | KY553147 |
| *Paradiplectanotrema klimpeli* | *Saurida tumbil* | Indonesia | MG763101 |
| *Platycephalotrema bassense* | *Platycephalus caeruleopunctatus* | Australia | MZ286653 |
| *Platycephalotrema platycephali* | *Platycephalus indicus* | Weihai, Shangdong, China | DQ157662 |
| *Protogyrodactylus alienus* | *Gerres filamentosus* | China | DQ157650 |
| *Protogyrodactylus hainanensis* | *Therapon jarbua* | Yangjiang, Guangdong, China | DQ157653 |
| *Pseudohaliotrema sphincteroporus* | *Siganus doliatus* | Australia | AF382058 |
| *Pseudempleurosoma haywardi* | *Johnius amblycephalus* | Indonesia | MF115715 |
| *Tetrancistrum labyrinthus* | *Siganus canaliculatus* | Oman | MN179332 |
| *Tetrancistrum indicum* | *Siganus canaliculatus* | Oman | MN179335 |
| *Xenoligophoroides cobitis* | *Gobius cobitis* | Black Sea, off Russia | MG194743 |
| Diplectanidae | | | |
| *Murraytrema pricei* * | *Nibea albiflora* | Panyu, China | DQ157672 |
| *Pseudorhabdosynochus lantauensis* * | *Epinephelus bruneus* | Huidong, China | AY553624 |
| *Pseudorhabdosynochus epinepheli* * | *Epinephelus bruneus* | Huidong, China | AY553622 |

* Species used as the outgroup.

## 3. Results

### 3.1. Order Dactylogyridea Bychowsky, 1937

- Dactylogyridae Bychowsky, 1933
- *Brotulella* n. gen. Cruces, Chero & Luque
- Diagnosis
- Body divided into cephalic region, trunk, peduncle, and haptor. Tegument thin, smooth. Head organs, cephalic lobes present; cephalic glands unicellular, anterior and posterolateral to pharynx. Eyespots absent; accessory chromatic granules absent. Mouth subterminal, midventral; pharynx muscular, glandular; oesophagus short, intestine bifurcated; intestinal caeca confluent posteriorly to testis, without diverticula. Common genital pore midventral, near level of intestinal bifurcation. Haptor armed with 14 hooks with ancyrocephaline distribution sensu Mizelle [15], 2 pairs of anchors, 2 haptoral bars, lacking haptoral reservoirs; anchors dissimilar, distal end of superficial and deep roots with stocking-shaped sheath; ventral bar bowed; dorsal bar with anterior and posterior broad process; hooks with undilated shanks and upright blunt thumb. Gonads tandem, intercaecal. Testis posterior to ovary. Vas deferens looping left intestinal caecum. Seminal vesicle almost sigmoid. Prostatic reservoirs two, with thick muscular walls. Male copulatory organ (MCO) tubular, sclerotized, with twisted distal end; base cylindrical; accessory piece delicate, membranous, articulated to shaft of MCO. Ovary U-shaped; oviduct, uterus not observed. Vagina sclerotized, emptying to seminal receptacle, vaginal aperture dextrolateral. Vitelline follicles dense, coextensive with intestinal caeca. Parasites of gills of marine fishes.
- Type species by original designation: *Brotulella laurafernandae* n. sp. from the gills of the Pacific bearded brotula *Brotula clarkae* Hubbs, 1944 (*Ophidiiformes: Ophidiidae*). Other species: *Brotulella luisahelenae* n. sp. from *B. clarkae*.
- Etymology: The genus name refers to the genus name of the fish host (*Brotula*). The diminutive *-ella* is appended to the genus name and should be treated as female.
- Remarks: *Brotulella* n. g. is characterized by the combination of the following features: (1) anchors with a stocking-shaped sheath associated with the distal end of superficial and deep roots, (2) tandem gonads, (3) a distally twisted MCO with a delicate membranous accessory piece articulated to the shaft of the MCO, (4) a U-shaped ovary, (5) an almost sigmoid seminal vesicle, and (6) two prostatic reservoirs with thick muscular walls. *Brotulella* n. g. most closely resembles species of the genus *Platycephalotrema* Kritsky & Nitta, 2019, which comprises nine species that infect the gill lamellae of the flathead fishes (Platycephalidae). Species of both genera share (1) tandem gonads, (2) a vas deferens looping left intestinal caecum, (3) two prostatic

reservoirs, (4) a sclerotized vagina, (5) dextral vaginal aperture, (6) seven pairs of hooks, (7) and tubular MCO with complex distal end. However, *Brotulella* n. g. differs from *Platycephalotrema* by its species having anchors unequal (anchors equal in *Platycephalotrema* spp.); a stocking-shaped sheath associated with the distal end of superficial and deep roots of the anchors (absent in *Platycephalotrema* spp.); a MCO with accessory piece (absent in *Platycephalotrema* spp.); hooks with upright blunt thumb (protruding blunt thumb in *Platycephalotrema* spp.); a U-shaped ovary (ovary entire in *Platycephalotrema* spp.); a short vaginal vestibule (large vaginal vestibule in *Platycephalotrema* spp.); prostatic reservoirs with thick muscular walls (prostatic reservoirs without thick muscular walls in *Platycephalotrema* spp.); and absence of eyespots (four eyespots in *Platycephalotrema* spp.).

- Species of *Brotulella* n. g. resembles species of *Ligophorus* Euzet & Suriano, 1977, which includes species parasitizing mullets (Mugilidae). Members of both genera share the following features: (1) gonads in tandem, (2) a tubular sclerotized and uncoiled MCO, (3) an ovary with U-shaped, and (4) a sclerotized vagina. However, species of these genera differ by having anchors with a stocking-shaped sheath associated with the distal end of superficial and deep roots (absent in *Ligophorus* spp.), a vas deferens looping the left intestinal caecum (vas deferens not looping the intestinal caecum in *Ligophorus* spp.), two prostatic reservoirs with thick muscular walls (one pyriform prostatic reservoir without thick muscular wall in *Ligophorus* spp.) and by having a ventral bar bowed (V-shaped ventral bar in *Ligophorus* spp.).

- Species of the new genus slightly resemble those of *Bravohollisia* Bychowsky & Nagibina, 1970 and *Caballeria* Bychowsky & Nagibina, 1970 by having tandem gonads, a tubular sclerotized and uncoiled MCO, and a bowed ventral bar. However, species of *Brotulella* n. g. differ from species of these genera by having a MCO with accessory piece (absent in *Bravohollisia* and *Caballeria* species) and by the absence of haptoral glands (present in *Bravohollisia* and *Caballeria* species). Species of the new genus slightly resembles species of *Mexicana* Caballero & Bravo-Hollis, 1959, parasitic on haemulid fishes, by having tandem gonads and an almost U-shaped ovary but differ from these species by having two prostatic reservoirs (one prostatic reservoir in *Mexicana* spp.), a testis not bipartite (principally bipartite posteriorly in *Mexicana* spp.) and absence of eyespots (four eyespots in *Mexicana* spp.).

- Finally, species of *Brotulella* n. gen. resemble species of the *Boegeriella* Mendoza-Palmero & Hsiao, 2020 and *Nanayella* Acosta, Mendoza-Palmero, da Silva & Scholz, 2019 by the shape of the ovary (U-shaped). However, species of *Brotulella* n. gen. differ from *Boegeriella* spp. by the morphology of the MCO (tubular with twisted distal end in *Brotulella* n. gen. vs coiled in *Boegeriella* spp.). Species of the new genus can be distinguished from species of *Nanayella* by having hooks of similar size (hook of dissimilar size in *Nanayella*).

### 3.1.1. *Brotulella laurafernandae* n. sp. Cruces, Chero & Luque

Body elongates (Figure 1), 486–702 (603, n = 13) long; greatest width 68–106 (89, n = 13) usually at level of testis. Cephalic region moderately broad; cephalic lobes moderately developed; cephalic glands bilateral, paired, at pre and postpharyngeal level. Pharynx spherical, greatest width 23–30 (27, n = 6). Peduncle short to elongated. Haptor subhexagonal, 59–84 (72, n = 13) long, 78–113 (100, n = 13) wide; group of inconspicuous secretory gland-cells lying on peduncle. Anchors with fine conspicuous alae; ventral anchor 59–63 (61, n = 7) long, with rounded deep root, robust superficial root, slightly arced shaft, elongated point, base 19–24 (21, n = 7) wide; dorsal anchor 65–69 (66, n = 7) long, with rounded deep root, elongated superficial root, slightly arced shaft, elongated point, base 20–23 (21, n = 7) wide. Ventral bar 44–57 (50, n = 11) long, rod-shaped, posteromedially bilobed, with expanded lateral ends. Dorsal bar 44–56 (51; n = 11) long, with mask-shaped, short anterior processes, curved posterior processes. Hooks 14, similar, 12–15 (13, n = 8) long, each with protruded obtuse thumb, uniform shank, and delicate point; filamentous hook

(FH) loop around shank length. MCO 36–46 (41, n = 12) long, tapered, accessory piece with medial and proximal expansions; base of MCO with cylinder-shaped. Testis large, intercaecal, ovate, not lobulated, 77–100 (89, n = 6) long, 42–58 (52, n = 6) wide; vas deferens dilate to form big seminal vesicle in left side of trunk, posterolateral to MCO; anterior prostatic reservoir ovate, dorsal to MCO, posterior prostatic reservoir with tadpole-shaped, dextrolateral to MCO. Ovary 32–55 (45, n = 6) long, 26–40 (34, n = 6) wide; oviduct, oötype and uterus not observed. Vaginal vestibule infundibuliform, sclerotized, lying horizontal on right side of body anterior to ovary; vaginal duct running posteriorly to join large sub-spherical seminal receptacle. Vitelline follicles throughout trunk, lateral fields of follicles confluent posterior to MCO and posterior to testis. Eggs not observed.

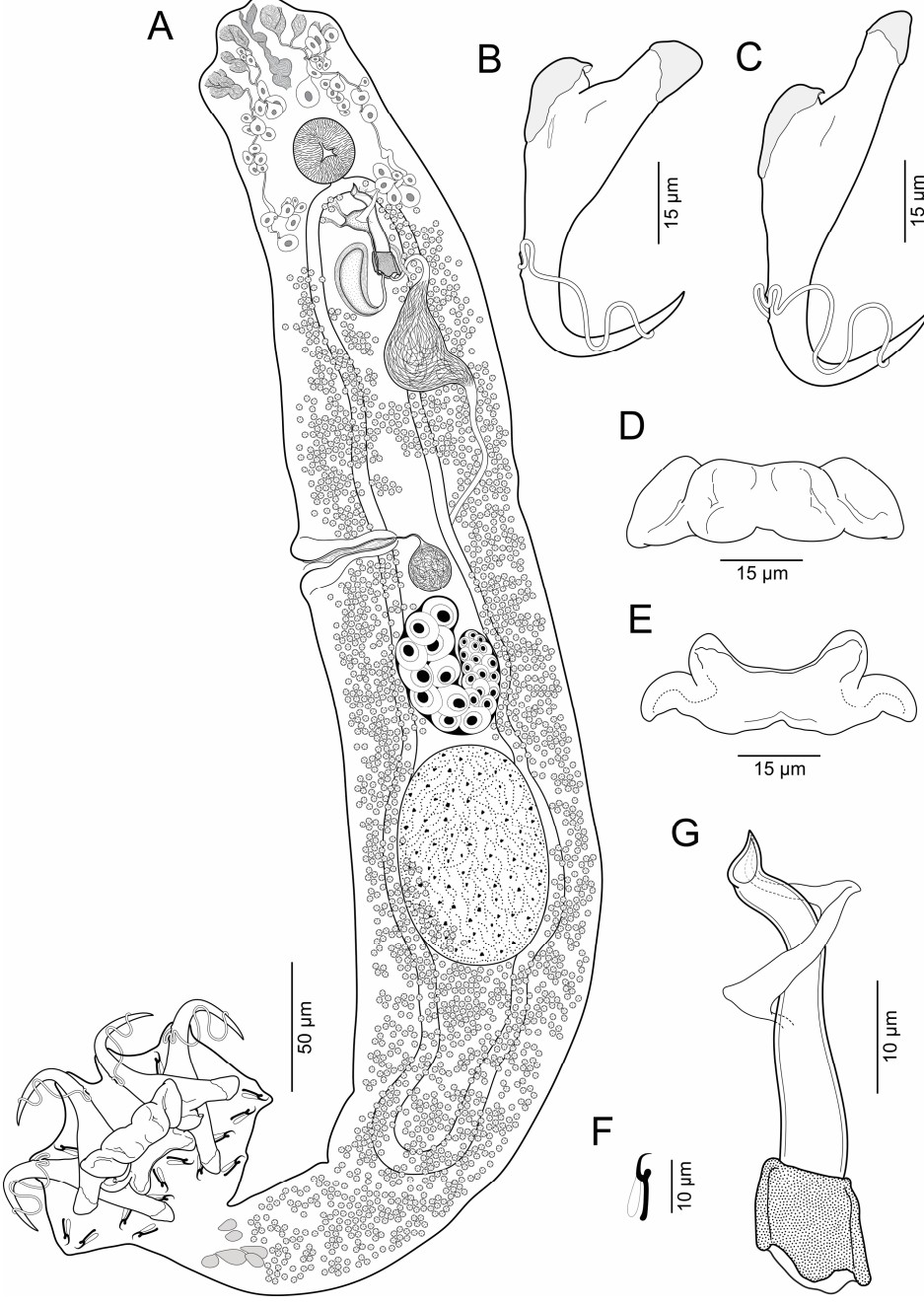

**Figure 1.** *Brotulella laurafernandae* n. sp. from the Pacific bearded brotula *Brotula clarkae* Hubbs, 1944 (*Ophidiiformes: Ophidiidae*). (**A**) Whole mount (composite, ventral view) (**B**) Ventral anchor (**C**) Dorsal anchor (**D**) Ventral bar (**E**) Dorsal bar (**F**) Hook (**G**) MCO.

- Type host: *Brotula clarkae* Hubbs, 1944 (*Ophidiiformes: Ophidiidae*), Pacific bearded brotula.
- Type locality: Puerto Pizarro resort (3°29′ S, 80°24′ W), Tumbes, Peru.
- Site in host: Gills.
- Type specimens: Holotype (MUSM-HEL 5132), 12 paratypes (MUSM-HEL 5133a–l), 1 hologenophore (MUSM 5133m).
- ZooBank registration: The Life Science Identifier (LSID) for *Brotulella laurafernandae* n. sp. is urn:lsid:zoobank.org:act:A1C5B22A-E8FF-4A70-B574-1B9A8B8082CC.
- Representative DNA sequence: Sequence was deposited in GenBank under the accession number OR860318 for the 28S rDNA with 720 bp.
- Etymology: The new species is named in honor of Laura Fernanda do Amarante Luque, daughter of the senior author.
- Remarks: *Brotulella laurafernandae* n. sp. is the type species of the genus.

3.1.2. *Brotulella luisahelenae* n. sp. Cruces, Chero & Luque

- Body elongates (Figure 2), 625–930 (769, n = 12) long; greatest width 111–153 (132, n = 12) usually at level of seminal vesicle. Cephalic region broad; cephalic lobes poorly developed; cephalic glands bilateral, paired at pre and postpharyngeal level. Pharynx spherical, in greatest width 29–44 (35, n = 8). Peduncle short to elongated. Haptor subquadrangular, 78–109 (94, n = 12) long, 120–165 (139, n = 12) wide; group of inconspicuous secretory gland-cells lying on peduncle. Anchors with fine conspicuous alae; ventral anchor 67–76 (70, n = 7) long, with rounded deep root, robust superficial root, slightly arced shaft, elongated point with furrow on external surface, base 22–26 (24, n = 7) wide; dorsal anchor 80–89 (88; n = 7) long, with almost rounded deep root, elongated superficial root, slightly arced shaft, elongated point with furrow on external surface, base 20–24 (22, n = 7) wide. Ventral bar 60–78 (69, n = 8) long, broadly V-shaped, with moderately enlarged lateral ends. Dorsal bar 56–73 (66, n = 8) long, with almost cat face-shaped, short anterior processes, oblique posterior processes. Hooks 14, similar, 12–15 (14, n = 7) long, each with protruded obtuse thumb, uniform shank, and delicate point; filamentous hook (FH) loop around shank length. MCO 88–107 (96, n = 12) long, tapered, accessory piece with proximal expansion; base of MCO with almost trapezium-shaped. Testis large, intercaecal, subtriangular, not lobulated, 91–129 (108, n = 5) long, 75–97 (88, n = 5) wide; vas deferens dilating to form big seminal vesicle in left side of trunk, posterolateral to MCO; ventral prostatic reservoir elongated, ventral to MCO, dorsal prostatic reservoir well-developed, dextrolateral to MCO. Ovary 42–61 (54, n = 5) long, 58–73 (65, n = 5) wide; oviduct, oötype and uterus not observed. Vaginal vestibule infundibuliform, sclerotized, lying horizontal on right side of body anterior to ovary; vaginal duct running posteriorly to join big subspherical seminal receptacle. Vitelline follicles throughout trunk, lateral fields of follicles confluent posterior to MCO and posterior to testis. Eggs not observed.
- Type host: *Brotula clarkae* Hubbs, 1944 (*Ophidiiformes: Ophidiidae*), Pacific bearded brotula.
- Type locality: Puerto Pizarro resort (3°29′ S, 80°24′ W), Tumbes, Peru.
- Site in host: Gills.
- Type specimens: Holotype (MUSM-HEL 5134), 12 paratypes (MUSM-HEL 5135a–k), 1 hologenophore (MUSM 5135l).
- ZooBank registration: The Life Science Identifier (LSID) for *Brotulella luisahelenae* n. sp. is urn:lsid:zoobank.org:act:6E75215A-471D-490E-809C-4968C7901A34.
- Representative DNA sequence: Sequence was deposited in GenBank under the accession number OR860321 for the 28S rRNA with 720 bp.
- Etymology: The new species is named in honor of Luisa Helena do Amarante Luque, daughter of the senior author.
- Remarks: *Brotulella luisahelenae* n. sp. can be distinguished from *Brotulella laurafernandae* n. sp. by its MCO, which have an almost trapezium-shaped base and a blanket-shaped membrane with proximal expansion. In addition, *B. luisahelenae* n. sp. is typified by having the points of the ventral anchors with a furrow on external surface.

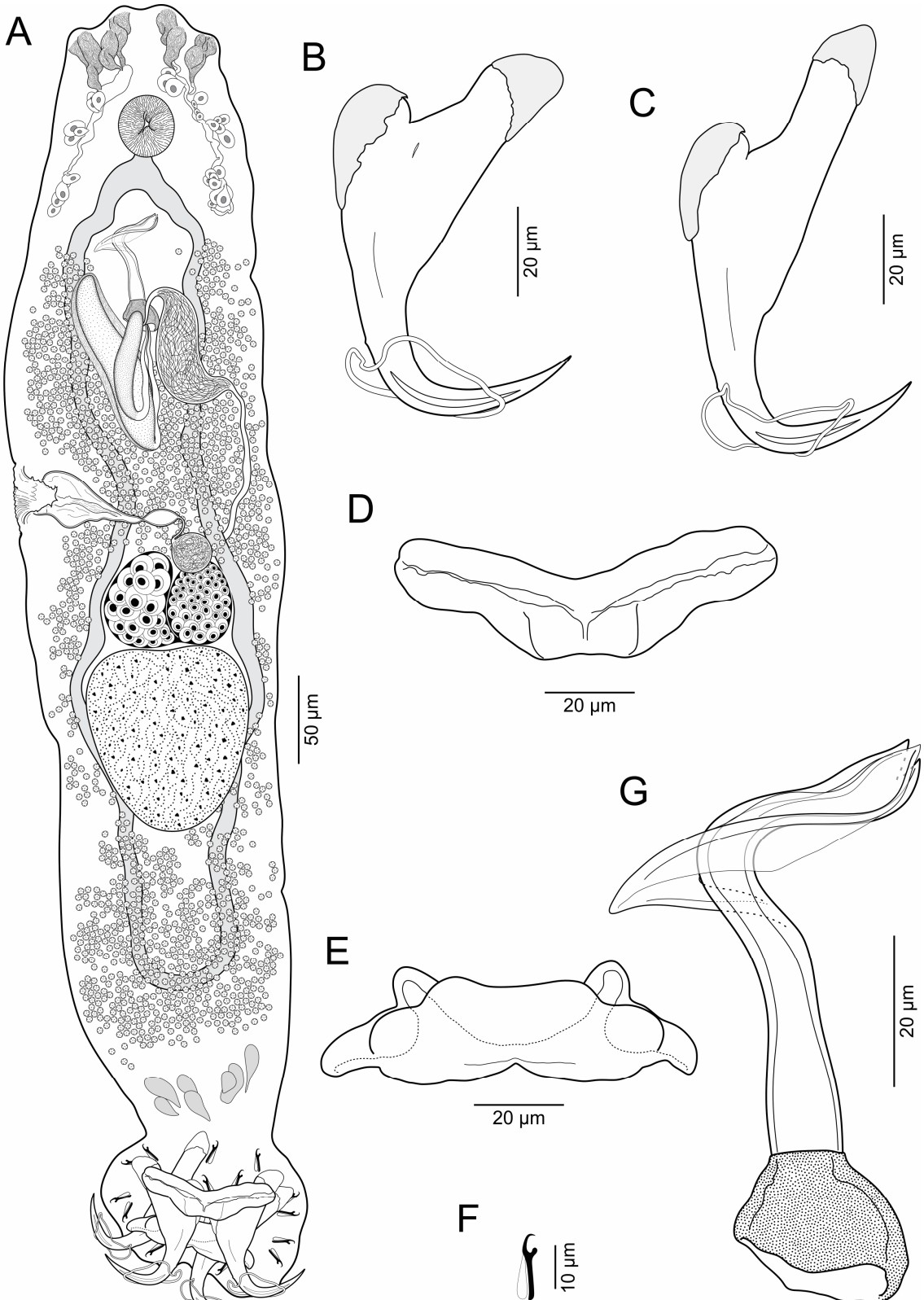

**Figure 2.** *Brotulella luisahelenae* n. sp. from the Pacific bearded brotula *Brotula clarkae* Hubbs, 1944 (*Ophidiiformes: Ophidiidae*). (**A**) Whole mount (composite, ventral view) (**B**) Ventral anchor (**C**) Dorsal anchor (**D**) Ventral bar (**E**) Dorsal bar (**F**) Hook (**G**) MCO.

### 3.2. Phylogenetic Relationships

The alignment of all analyzed species comprised 769 characters, with 229 being constant and 478 being parsimony-informative variables. The Bayesian analyses yielded a mean estimated marginal likelihood of −11,962.2929, accompanied by a median value of −11,961.93. The effective sample size for all parameters exceeded 100, indicating a substantial number of effectively independent samples. Moreover, this sample size was considerably larger for the majority of parameters, underscoring the robust nature of the samples. ML and BI analyses generated phylogenetic trees with similar general topology, with little variation in node support values (Figure 3). The sequences obtained from the new species differed in 10 bases. The smallest distance was found between the sequences of *Brotulella laurafernandae* n. sp. and *Brotulella luisahelenae* n. sp. (0.01%). The distance between the two new species and the sister genera are: 0.12% for *Platycephalotrema bassense* and 0.15% for *Haliotrema johnstoni* and *P. platycephali*. The species *Brotulella laurafernandae* n. sp. and *Brotulella luisahelenae* n. sp. formed a strongly supported clade with Bayesian posterior probabilities (BPP = 0.97) and ML (aLRT = 0.99; ML-BP = 1000) values. This clade also constituted a moderately supported monophyletic group alongside *Platycephalotrema bassense*, *Haliotrema johnstoni* and *P. platycephali* (BPP = 0.97; aLRT = 0.66; ML-BP = 603) (Figure 3). These species are positioned within a significantly supported major clade referred to as the '*Haliotrema*' group as per Kmentová et al. [33]). This clade encompasses other species such as those belonging to *Bravohollisia* Bychowsky & Nagibina, 1970, *Caballeria* Bychowsky & Nagibina, 1970, *Lethrinitrema* Lim & Justine, 2011, *Pseudohaliotrema* Yamaguti, 1953 and *Tetrancistrum* Goto & Kikuchi, 1917.

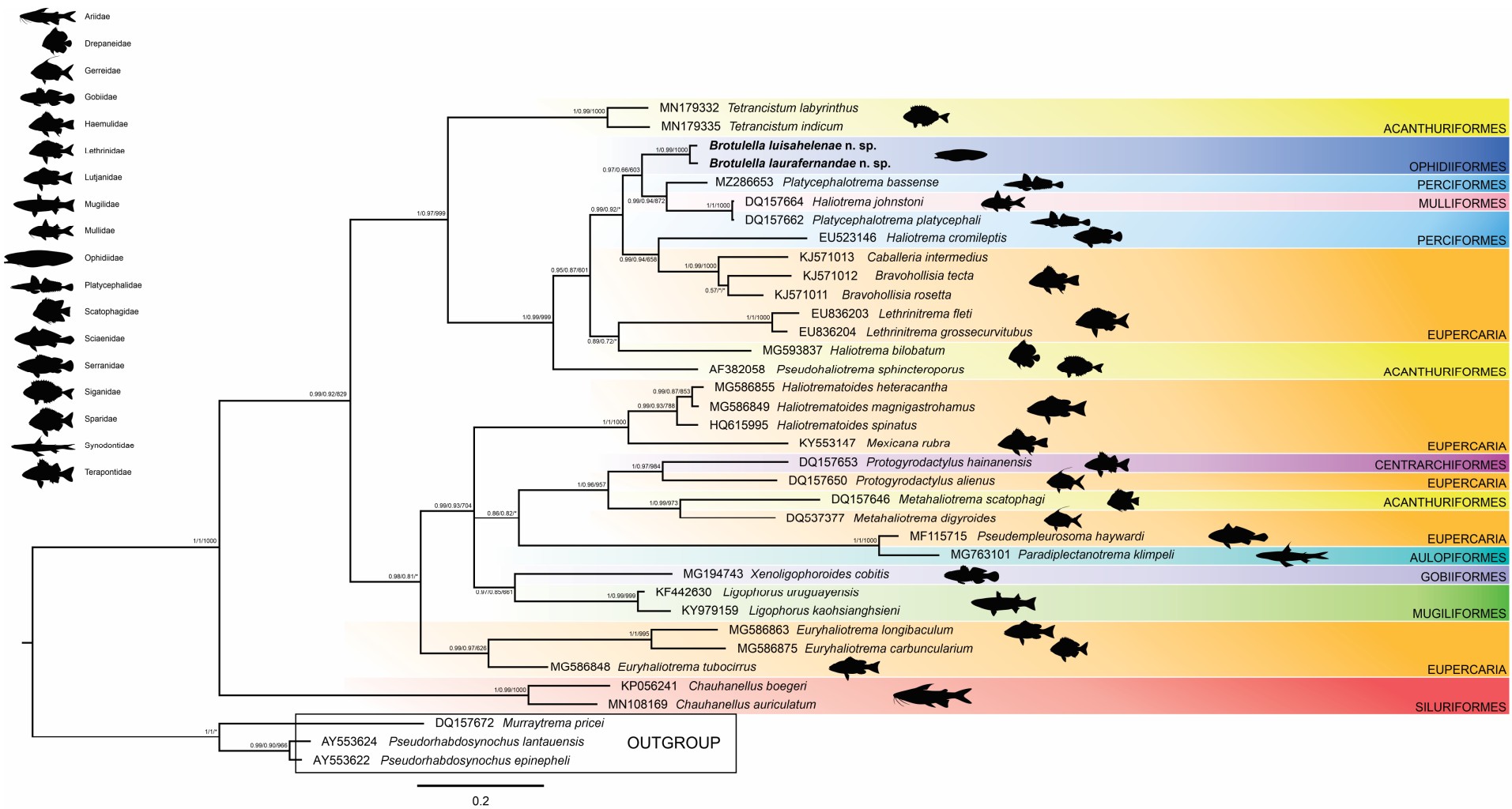

**Figure 3.** Phylogenetic tree based on 28S region for *B. laurafernandae* n. sp. and *B. luisahelenae* n. sp. (Dactylogyridae) inferred by using the Bayesian inference (BI) and maximum likelihood (ML) methods (aLRT and bootstrap replicates). The nodal support is described at the left by bayesean posterior probability, aRLT and bootstrap replicates to each node represented. The scale bar represents the number of substitutions per site. The symbol of one asterisk (*) indicates low nodal support.

## 4. Discussion

In the present study, *Brotulella* n. gen. is proposed to accommodate two marine dactylogyrids, *B. laurafernandae* n. sp. and *B. luisahelenae* n. sp., infecting the gill filaments of the Pacific bearded brotula in the Southeast Pacific off Peru, based on the presence of a combination of morphological characteristics which include anchors with a stocking-shaped sheath associated with the distal end of superficial and deep roots, tandem gonads, a distally twisted MCO with an accessory piece articulated to the shaft of the MCO, a U-shaped ovary, and two prostatic reservoirs with thick muscular walls. Moreover, the phylogenetic position of the species of *Brotulella* n. gen. support and justify the erection of the new genus. Our partial sequences of the 28S rDNA of both new species of *Brotulella* n. gen. from the Southeast Pacific Ocean cluster within a strongly supported clade that includes *Platycephalotrema bassense* (Hughes, 1928) Kritsky & Nitta, 2019 from Australia, along with *Haliotrema johnstoni* Bychowsky & Nagibina, 1970, and *P. platycephali* (Yin & Sproston, 1948) Kritsky & Nitta, 2019 from China. Species of *Brotulella* n. gen. and the two *Platycephalotrema* species share some morphological characteristics such as two seminal vesicles and tandem gonads.

*Haliotrema johnstoni*, a dactylogyrid infecting the dark-barred goatfish *Upeneus luzonius* Jordan & Seale, 1907 (Mullidae), is nested within *Platycephalotrema*. The genetic similarity between *H. johnstoni* and the two *Platycephalotrema* species observed in the present analysis appears to be consistent with the morphological data, i.e., a dorsal bar with bifurcating ends in *H. johnstoni* and *Platycephalotrema* species, as mentioned by Kmentová et al. [33]. Unfortunately, we were unable to access the type material of *H. johnstoni* to corroborate this hypothesis. Furthermore, as mentioned by Kmentová et al. [33] the presence of a MCO with an accessory piece and the fact that *H. johnstoni* infect a different host repertoire provide evidence that the relationship between *H. johnstoni* and *Platycephalotrema* species warrants further investigation.

Brotulas, *Brotula* spp. (Ophidiidae), are principally benthopelagic fishes that occur in the Atlantic and Pacific Oceans [2]. Previously, only one of the six valid species of *Brotula* [2] has been reported as host for monogeneans [34]. *Haliotrema brotulae* Yamaguti, 1968 and *H. spiculare* are the only species of monogeneans described from these hosts [34]. Both species have some morphological similarities with the new species described here, i.e., a distally twisted MCO, lacking accessory piece, an almost sigmoid seminal vesicle, and two prostatic reservoirs, these species could eventually be removed from *Haliotrema* Johnston & Tiegs, 1922 and transferred to the new genus. However, our endeavor to assess the type material was thwarted by complications related to gaining access to the collection, consequently impeding an exhaustive investigation. Additionally, there are no available sequences of the 28S rRNA gene of *H. brotulae* and *H. spiculare* to test their phylogenetic relationship with species of *Brotulella* n. gen. Hence, a meticulous study based on the examination of both type and novel specimens of *H. brotulae* and *H. spiculare*, in conjunction with genetic data, is required to confirm their taxonomic classification.

The taxonomy of monogeneans has long been a subject of scientific interest, and the utilization of alternative tools to address taxonomic uncertainties enhances our understanding of monogenean diversity [35–37]. The application of molecular tools has proven invaluable in resolving taxonomic ambiguities within monogeneans [38,39]. Through the analysis of DNA sequences and the construction of phylogenetic trees, researchers have gained insights into the evolutionary history and genetic diversity of monogenean species [40–43]. Additionally, comparative genomics has provided a deeper understanding of the genomic characteristics and evolutionary adaptations of these parasites [39]. Furthermore, the integration of alternative tools, such as transcriptomics and proteomics, can offer valuable insights into the functional genomics of monogeneans [39].

The knowledge of the marine monogenean parasite fauna of Peruvian fishes has increased remarkably in the last years [4,5]. Most of these taxonomic studies were solely based on morphological data. Only four studies provided molecular data on monogeneans of the Diclidophoridae (3 species), Hexabothriidae (2 species) and Monocotylidae (1 species)

families [44]. Thus, the present study provides the first sequence of the partial rRNA gene 28S region of marine dactylogyrids from Peru.

Nineteen marine dactylogyrid monogenean species from the genera *Bicentenariella* Cruces, Chero, Sáez & Luque, 2021 (5 spp.), *Haliotrema* Johnston & Tiegs, 1922 (3 spp.), *Haliotrematoides* Kritsky, Yang & Sun, 2009 (1 species), *Euryhaliotrema* (4 spp.) Kritsky & Boeger, 2002, *Mexicana* Caballero & Bravo-Hollis, 1959 (1 species), *Pronotogrammella* Cruces, Chero, Sáez & Luque, 2020 (3 spp.), and *Tylosuricola* Unnithan, 1964 (1 species) have been described or reported infecting the gills of eleven fish species from Peru [5]. From these, three species are known for parasitizing the gills of fishes captured in central Peru [45]. The other sixteen species of dactylogyrids infect fishes in northern Peru. The two new species described here increase the number of dactylogyrid species that infect fishes from Peru to 21.

## 5. Conclusions

In this study, a new genus, *Brotulella* n. gen., is proposed to accommodate two newly discovered species of monogenean parasites found on the gills of the Pacific bearded brotula, *Brotula clarkae*. These species in the new genus are distinguished by unique morphological features, including specific characteristics of their anchors, gonads, male copulatory organ, ovary, seminal vesicle, and prostatic reservoirs. The phylogenetic analysis based on 28S ribosomal DNA sequences supports the establishment of the new genus and provides insights into the biodiversity of these parasites in the Southeastern Pacific Ocean.

**Author Contributions:** C.L.C., J.D.C. and J.L.L. conceived and designed the study; J.D.C. and C.L.C. carried out the field work; C.L.C., R.S. (Raquel Simões) and A.M.J. performed molecular analyses. Additional analyses were performed by C.L.C., J.D.C., R.S. (Raquel Simões), R.S. (Ruperto Severino) and J.L.L.; J.D.C. and C.L.C. wrote the manuscript. All authors have read and agreed to the published version of the manuscript.

**Funding:** C.L.C. was supported by a student fellowship from the Coordenação de Aperfeiçoamento de Pessoal do Ensino Superior, Brazil (CAPES)—Finance Code 001. J.L.L. was supported by a Researcher fellowship from the Conselho Nacional de Desenvolvimento Científico e Tecnológico, Brazil (CNPq).

**Institutional Review Board Statement:** This study did not consider experiments with live animals. All fishes were obtained from commercial catches, and none of the species are subject to conservation measures.

**Data Availability Statement:** Data are contained within the article.

**Acknowledgments:** The authors are grateful to the following people who helped with the collection of fishes in Peru: Milagros K. Carrillo, Alexander Reyes, and Cynthia E. Rodríguez, all from the National University Federico Villarreal (UNFV).

**Conflicts of Interest:** The authors declare no conflict of interest.

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
