# Peer review of "Proposal of Brotulella n. gen. for Monogeneans from the Gills of the Pacific Bearded Brotula Brotula clarkae Hubbs, 1944 (Ophidiiformes: Ophidiidae) Based on Morphological and Molecular Evidence"

_fishes, doi:10.3390/fishes8120588_

Round 1

Reviewer 1 Report

Comments and Suggestions for Authors

Nice MS. 

Perhaps a revolutionary change in the systematics of Monogenea could be mentioned in the article (Brabec et al. 2023, DOI 10.1016/cub202308.064), but it is only an idea…

Author Response

Response to Reviewer 1 Comments

Nice MS.

Perhaps a revolutionary change in the systematics of Monogenea could be mentioned in the article (Brabec et al. 2023, DOI 10.1016/cub202308.064), but it is only an idea…

-Ok. It was made.

Reviewer 2 Report

Comments and Suggestions for Authors

This manuscript proposes a new genus and two new species. The manuscript is succinct and focussed, and mostly well-written, although there are numerous simple errors which ought to have been corrected prior to submission. The illustrations are excellent and the descriptions are highly detailed. The combined morphological and molecular evidence is compelling. I have listed numerous minor corrections and suggestions below, and have few larger concerns which I consider amount only to ‘minor corrections required’:

The spelling of the new genus name is not consistent throughout the manuscript. In particular, it is spelled Brotullela where it is actually proposed and diagnosed, but mostly Brotulella elsewhere.

I am not an expert on nomenclature. However, the new genus name seemed misleading to me and perhaps would be more appropriate for a fish genus than a worm genus. Brotulella seems to mean "little brotula" and so evokes a fish smaller and similar to Brotula spp., not a worm parasitising a brotula. I encourage the authors to reconsider but leave it to their discretion.

There are no details of how the molecular alignment was performed in the Methods. 

The two species are distinguished only on the basis of morphology, despite the novel molecular data. Nowhere in the results is it made clear how different the species were in the 28S data, nor is that explicitly considered as part of the justification in delineating the two species, despite claiming a combined approach.

The descriptions are thoroughly detailed. However, there are some issues with description formatting and style throughout. First, and most importantly, the use of commas vs semi-colons vs new sentences is inconsistent and confusing in several places. My understanding is that each major organ/character gets a new sentence, whereas semi-colons are used to separate suborgans/subcharacters of the main organ, and commas are used to list adjectives, metrics etc relating to the organ or suborgan. In this manuscript, semi-colons are sometimes (but not always) used to separate related major organs, e.g. mouth, pharynx, gut, rather than beginning a new sentence for each - I find this permissible but consistency is lacking, e.g. testis and seminal vesicle separated by a semi-colon but then a new sentence for the MCO. In other places, semi-colons have been used to separate metrics rather than commas. Second, sentences and clauses (i.e. after a semi-colon) should begin with the subject, the noun, e.g. hooks 14 vs fourteen hooks. Third, use the simple present tense vs the present participle (i.e. avoid the -ing form of verbs). Please thoroughly and carefully edit the diagnosis and descriptions.

Title: Perhaps some higher taxonomy for the new genus is desirable? Especially as Brotulella sounds like a fish rather than a worm.

Abstract

20: I suggest simplify "the Puerto Pizarro locality" to "Puerto Pizarro".

26: There were two analyses, ML and BI.

27: "sequences"

Introduction

40: "harbors", plural, because the subject is "Peru", not "countries".

41: From what I could tell, reference [1] does not appear to support the statement being made. The statement is about fish richness in Peru, especially for bony fishes. The referenced work is a DNA barcoding survey of landed fishes at a local fishery; it includes the richness of sharks in Peruvian waters (referring to another study), but not bony fishes. 

41: The first part of this sentence "Regarding to its marine ichthyology diversity" is unnecessary and awkward.

43: What is the context here? One of the richest marine ecosystems in Peru? Worldwide?

44: Replace "showing" with "show", or better "have found".

47: Remove "of" before "belonging".

47-49: I do not think it appropriate to conclude in the Introduction that the taxa considered warrant proposal of a new genus and new species. This finding is the core result in the manuscript - save it for the Results section.

Methods

59: Replace "while" with "and" or "whereas".

79: I suggest replace "The samples of DNA" with "Genomic DNA".

81: No, only the target region was amplified. I suggest replace "This DNA" with "Partial 28S" and finish the sentence at [19, 20, 21].

83: "This DNA was amplified... for amplification" does not make sense.

84: The gene is rDNA, the product is rRNA. So "28S rDNA" or perhaps "28S rRNA gene", but I do not think "28S rDNA gene" is correct.

86: Extra period after 72 degrees C.

89: I suggest replace "contig" with "contiguous sequences"

90: Genuine ambiguities should not be eliminated, the phrasing could suggest here that the authors set out to "fix" rather than check their data.

90: Remove the last part of the sentence "to provide consensus sequences", already said that.

91: "were"

91-92: I think start this part about the analyses by making it clear that they were reconstructed with both ML and BI. 

93-94: Which model of nucleotide evolution was used?

98-99: Replace "[28]. The 28S region Bayesian analyses were performed using a single" with "with the". No need to mention 28S again here, nor to repeat that it was a Bayesian model. Above, for ML, SMS was used to select the model of evolution, here no justification is made for the choice of model.

103: I suggest replace "used" with "included". There is no mention of construction of alignments. I think this is required, and intuitively ought to come after sequencing and before the analyses. How long was the final alignment? How were ambiguously aligned characters treated?

103: "analyses", plural, ML and BI.

104: "as well as".

Table 1

105: I find the use of "marine" here odd and confusing. Are there freshwater dactylogyrids that were not included? If so, why? If not, why specify here that the included data are from marine species?

Results

113: Is it Brotullela or Brotulella??

112-113: I think "Diagnosis" should be a subheading under "Brotullela n. gen.". At the moment, "Diagnosis" is a subheading under "Dactylogyridae", but it is the genus being diagnosed here, not the family.

114: I think "surface" is implicit?

117-118: Replace "bifurcating to form two intestinal caeca;" with "intestine bifurcated;". The use of "two" is redundant. The authors don't specify what (the intestine) bifurcates.

118: Replace "lacking" with "lack" - use the simple present vs present participle. Actually, perhaps "without" is better here?

120: Testis appears to be posterior to ovary in both illustrations, not dorsal?

120: The use of new sentences vs semi-colons seems inconsistent here, for the male bits. Why "testis; seminal vesicle", but then a new sentence for prostatic reservoirs, and a new line again for MCO?

120: Sentence should start with subject: "Prostatic reservoirs two, with thick muscular walls".

122: Here I find the use of semi-colons vs commas inconsistent: why is "distal end" included with MCO via a comma but the base (of the MCO presumably) gets a new clause? Add subject should come first, so "base cylindrical". There is word for cylinder-shaped! Likewise, why a new clause for the accessory piece? If it does deserve a new clause, then "accessory piece delicate, membranous, articulated to shaft of MCO."

123-124: I would think vagina should come before vaginal aperture, the aperture is a subcharacter.

127: Here there is a new sentence for anchors, but they already featured in the previous sentence.

128: Likewise, why a new sentence vs semincolon for bars? 

129: "fishes"?

130: Best to add "by original designation" to be explicit.

133: So the etymology is "little brotula"... but its a genus of worm??

139-140: This sentence does not make sense, either the genus is similar to another genus concept or the species of the genus resemble those of another genus, but a genus cannot resemble another genus, because a genus is a concept, not a thing.

141: "includes", but perhaps "comprises" is more precise?

153: As above, Brotulella is similar to, or its species resemble those of, but it cannot resemble another genus, as a concept has no semblance.

156: Replace "both" with "species of these", and remove "from each other".

161; Replace "resembles to" with "resemble those of"

163: "bowed ventral bar", adjective before noun (except in description).

166: Same problem as above.

179: I believe fusiform is a 3D shape, yet I think here it is being used in 2D. I suspect the worms are quite flat and therefore not fusiform.

179: Not sure a semi-colon is necessary within the parentheses, here and throughout. Would a comma not be more appropriate? Check throughout descriptions.

179: Why is width separated from length by a semi-colon? Semi-colons can be used to separate related or subcharacters from the main subject character, but width is a metric, not a subcharacter.

181: Start with subject: "cephalic glands bilateral, paired.."

182: Remove "in".

182: In the genus diagnosis, the haptor came towards the end, here it comes after the pharynx. I suppose the genus diagnosis and species description need not necessarily be consistent in order, but I do not see here why the shouldn't or couldn't be?

190: Subject first: "Hooks 14, similar..."

192: Ambiguous. Does "about" here mean the loop is similar in length to the shank or that the FH loops around the shank's length?

194: Replace dilating with dilate. Use simple present form vs present participle.

195: large vs big?

210: Should it not be sequence, singular? The methods state only one sequence was generated per species?

211: The data are rDNA, not rRNA.

219: Italics missing. Many of the comments for the description of the above species apply here and are not repeated.

266-268: Would this be better placed in Methods?

Discussion

301: Confirms is a strong word. A phylogenetic analysis cannot confirm proposal of a new genus is warranted, because a genus is a subjective hypothesis, and because not all taxa are sequenced. The analyses can be used to support and justify the proposal, but not confirm it.

302: The sentence is incomplete.

294-308: So far all this repeats from the Results, except the incomplete sentence justifying the new genus.

308-317: This part feels like it should be a separate paragraph, and it feels to earlier in the discussion to already not be discussion Brotulella, the focus.

319: I think "Previously," vs "To date", to make clearer that this is the second report and not the first from a Brotula.

320: "has"

331: I don't think "status" is the right term here. Perhaps something like "classification"? And neither examining the type-material nor sequencing the worms is "required" to transfer the species to the new genus, although these may be desirable.

333: "remarkably increased" and "increased remarkably" are not equivalent, I suspect the authors mean the latter here?

334: "based employed"?? "and barely,"??

Comments on the Quality of English Language

The writing and English are mostly to a high standard, although there are many simple errors throughout, see list above.

Author Response

Response to Reviewer 2 Comments

This manuscript proposes a new genus and two new species. The manuscript is succinct and focussed, and mostly well-written, although there are numerous simple errors which ought to have been corrected prior to submission. The illustrations are excellent and the descriptions are highly detailed. The combined morphological and molecular evidence is compelling. I have listed numerous minor corrections and suggestions below, and have few larger concerns which I consider amount only to ‘minor corrections required’:

The spelling of the new genus name is not consistent throughout the manuscript. In particular, it is spelled Brotullela where it is actually proposed and diagnosed, but mostly Brotulella elsewhere.

-Ok. It was corrected. The name correct is Brotulella.

I am not an expert on nomenclature. However, the new genus name seemed misleading to me and perhaps would be more appropriate for a fish genus than a worm genus. Brotulella seems to mean "little brotula" and so evokes a fish smaller and similar to Brotula spp., not a worm parasitising a brotula. I encourage the authors to reconsider but leave it to their discretion.

-Dear reviewer,

Thank you very much for your comment and for taking the time to analyze the nomenclature of the genus. We genuinely appreciate your perspective on the proposed genus name. However, after careful consideration of various factors, including the criteria followed by other studies in the field, we have chosen to maintain the genus name as it stands.

 There are no details of how the molecular alignment was performed in the Methods. 

-Ok. It was added.

The two species are distinguished only on the basis of morphology, despite the novel molecular data. Nowhere in the results is it made clear how different the species were in the 28S data, nor is that explicitly considered as part of the justification in delineating the two species, despite claiming a combined approach.

-Ok. It was added.

The descriptions are thoroughly detailed. However, there are some issues with description formatting and style throughout. First, and most importantly, the use of commas vs semi-colons vs new sentences is inconsistent and confusing in several places. My understanding is that each major organ/character gets a new sentence, whereas semi-colons are used to separate suborgans/subcharacters of the main organ, and commas are used to list adjectives, metrics etc relating to the organ or suborgan. In this manuscript, semi-colons are sometimes (but not always) used to separate related major organs, e.g. mouth, pharynx, gut, rather than beginning a new sentence for each - I find this permissible but consistency is lacking, e.g. testis and seminal vesicle separated by a semi-colon but then a new sentence for the MCO. In other places, semi-colons have been used to separate metrics rather than commas. Second, sentences and clauses (i.e. after a semi-colon) should begin with the subject, the noun, e.g. hooks 14 vs fourteen hooks. Third, use the simple present tense vs the present participle (i.e. avoid the -ing form of verbs). Please thoroughly and carefully edit the diagnosis and descriptions.

-Ok it was corrected.

Title: Perhaps some higher taxonomy for the new genus is desirable? Especially as Brotulella sounds like a fish rather than a worm.

- Thank you very much for your comment and for taking the time to analyze the nomenclature of the genus. We genuinely appreciate your perspective on the proposed genus name. However, after careful consideration of various factors, including the criteria followed by other studies in the field, we have chosen to maintain the genus name as it stands.

Abstract

20: I suggest simplify "the Puerto Pizarro locality" to "Puerto Pizarro".

-Ok it was made.

26: There were two analyses, ML and BI.

-Ok. It was added.

27: "sequences"

-Ok. It was corrected.

Introduction

40: "harbors", plural, because the subject is "Peru", not "countries".

-Ok. It was corrected.

41: From what I could tell, reference [1] does not appear to support the statement being made. The statement is about fish richness in Peru, especially for bony fishes. The referenced work is a DNA barcoding survey of landed fishes at a local fishery; it includes the richness of sharks in Peruvian waters (referring to another study), but not bony fishes. 

-Ok. It was corrected.

41: The first part of this sentence "Regarding to its marine ichthyology diversity" is unnecessary and awkward.

-Ok. It was corrected.

43: What is the context here? One of the richest marine ecosystems in Peru? Worldwide?

-Ok. It was corrected.

44: Replace "showing" with "show", or better "have found".

-Ok. It was corrected.

47: Remove "of" before "belonging".

-Ok. It was made.

47-49: I do not think it appropriate to conclude in the Introduction that the taxa considered warrant proposal of a new genus and new species. This finding is the core result in the manuscript - save it for the Results section.

 -Ok. It was made.

Methods

59: Replace "while" with "and" or "whereas".

-Ok. It was made.

79: I suggest replace "The samples of DNA" with "Genomic DNA".

-Ok. It was made.

81: No, only the target region was amplified. I suggest replace "This DNA" with "Partial 28S" and finish the sentence at [19, 20, 21].

-Ok. It was made.

83: "This DNA was amplified... for amplification" does not make sense.

-Ok. It was made.

84: The gene is rDNA, the product is rRNA. So "28S rDNA" or perhaps "28S rRNA gene", but I do not think "28S rDNA gene" is correct.

-Ok. It was made.

86: Extra period after 72 degrees C.

-Ok. It was made.

89: I suggest replace "contig" with "contiguous sequences"

-Ok. It was made.

90: Genuine ambiguities should not be eliminated, the phrasing could suggest here that the authors set out to "fix" rather than check their data.

-Ok. It was corrected.

90: Remove the last part of the sentence "to provide consensus sequences", already said that.

-Ok. It was made.

91: "were"

-Ok. It was made.

91-92: I think start this part about the analyses by making it clear that they were reconstructed with both ML and BI. 

-Ok. It was made.

93-94: Which model of nucleotide evolution was used?

-Ok. It was added.

98-99: Replace "[28]. The 28S region Bayesian analyses were performed using a single" with "with the". No need to mention 28S again here, nor to repeat that it was a Bayesian model. Above, for ML, SMS was used to select the model of evolution, here no justification is made for the choice of model.

-Ok. It was made.

103: I suggest replace "used" with "included". There is no mention of construction of alignments. I think this is required, and intuitively ought to come after sequencing and before the analyses. How long was the final alignment? How were ambiguously aligned characters treated?

-Ok. It was made.

103: "analyses", plural, ML and BI.

-Ok. It was made.

104: "as well as".

-Ok. It was made.

Table 1

105: I find the use of "marine" here odd and confusing. Are there freshwater dactylogyrids that were not included? If so, why? If not, why specify here that the included data are from marine species?

-Ok. It was changed.

Results

113: Is it Brotullela or Brotulella??

-Ok. It was corrected.

112-113: I think "Diagnosis" should be a subheading under "Brotullela n. gen.". At the moment, "Diagnosis" is a subheading under "Dactylogyridae", but it is the genus being diagnosed here, not the family.

-Ok. It was made.

114: I think "surface" is implicit?

-Ok. It was made.

117-118: Replace "bifurcating to form two intestinal caeca;" with "intestine bifurcated;". The use of "two" is redundant. The authors don't specify what (the intestine) bifurcates.

-Ok. It was made.

118: Replace "lacking" with "lack" - use the simple present vs present participle. Actually, perhaps "without" is better here?

-Ok. It was made.

120: Testis appears to be posterior to ovary in both illustrations, not dorsal?

-Ok. It was made.

120: The use of new sentences vs semi-colons seems inconsistent here, for the male bits. Why "testis; seminal vesicle", but then a new sentence for prostatic reservoirs, and a new line again for MCO?

-Ok. It was made.

120: Sentence should start with subject: "Prostatic reservoirs two, with thick muscular walls".

-Ok. It was made.

122: Here I find the use of semi-colons vs commas inconsistent: why is "distal end" included with MCO via a comma but the base (of the MCO presumably) gets a new clause? Add subject should come first, so "base cylindrical". There is word for cylinder-shaped! Likewise, why a new clause for the accessory piece? If it does deserve a new clause, then "accessory piece delicate, membranous, articulated to shaft of MCO."

-Ok. It was corrected.

123-124: I would think vagina should come before vaginal aperture, the aperture is a subcharacter.

-Ok. It was made.

127: Here there is a new sentence for anchors, but they already featured in the previous sentence.

-Ok. It was corrected.

128: Likewise, why a new sentence vs semincolon for bars? 

-Ok. It was corrected.

129: "fishes"?

-Ok. It was add.

130: Best to add "by original designation" to be explicit.

-Ok. It was made.

133: So the etymology is "little brotula"... but its a genus of worm??

-Thank you very much for your comment and for taking the time to analyze the nomenclature of the genus. We genuinely appreciate your perspective on the proposed genus name. However, after careful consideration of various factors, including the criteria followed by other studies in the field, we have chosen to maintain the genus name as it stands.

139-140: This sentence does not make sense, either the genus is similar to another genus concept or the species of the genus resemble those of another genus, but a genus cannot resemble another genus, because a genus is a concept, not a thing.

-Ok. It was corrected.

141: "includes", but perhaps "comprises" is more precise?

-Ok. It was made.

153: As above, Brotulella is similar to, or its species resemble those of, but it cannot resemble another genus, as a concept has no semblance.

-Ok. It was corrected.

156: Replace "both" with "species of these", and remove "from each other".

-Ok. It was made.

161; Replace "resembles to" with "resemble those of"

-Ok. It was made.

163: "bowed ventral bar", adjective before noun (except in description).

-Ok. It was made.

166: Same problem as above.

-Ok. It was made.

179: I believe fusiform is a 3D shape, yet I think here it is being used in 2D. I suspect the worms are quite flat and therefore not fusiform.

-Ok. It was made.

179: Not sure a semi-colon is necessary within the parentheses, here and throughout. Would a comma not be more appropriate? Check throughout descriptions.

-Ok. It was corrected.

179: Why is width separated from length by a semi-colon? Semi-colons can be used to separate related or subcharacters from the main subject character, but width is a metric, not a subcharacter.

-Ok. It was corrected.

181: Start with subject: "cephalic glands bilateral, paired.."

-Ok. It was made.

182: Remove "in".

-Ok. It was made.

182: In the genus diagnosis, the haptor came towards the end, here it comes after the pharynx. I suppose the genus diagnosis and species description need not necessarily be consistent in order, but I do not see here why the shouldn't or couldn't be?

-Ok. It was corrected.

190: Subject first: "Hooks 14, similar..."

-Ok. It was corrected.

192: Ambiguous. Does "about" here mean the loop is similar in length to the shank or that the FH loops around the shank's length?

-Ok. It was corrected.

194: Replace dilating with dilate. Use simple present form vs present participle.

-Ok. It was made.

195: large vs big?

-Ok. It was corrected.

210: Should it not be sequence, singular? The methods state only one sequence was generated per species?

-Ok. It was corrected.

211: The data are rDNA, not rRNA.

-Ok. It was corrected.

219: Italics missing. Many of the comments for the description of the above species apply here and are not repeated.

-Ok. It was made.

266-268: Would this be better placed in Methods?

-Ok. It was made.

Discussion

301: Confirms is a strong word. A phylogenetic analysis cannot confirm proposal of a new genus is warranted, because a genus is a subjective hypothesis, and because not all taxa are sequenced. The analyses can be used to support and justify the proposal, but not confirm it.

-Ok. It was corrected.

302: The sentence is incomplete.

-Ok. It was corrected.

294-308: So far all this repeats from the Results, except the incomplete sentence justifying the new genus.

-Ok. It was made.

308-317: This part feels like it should be a separate paragraph, and it feels to earlier in the discussion to already not be discussion Brotulella, the focus.

-Ok. It was corrected.

319: I think "Previously," vs "To date", to make clearer that this is the second report and not the first from a Brotula.

-Ok. It was made.

320: "has"

-Ok. It was made.

331: I don't think "status" is the right term here. Perhaps something like "classification"? And neither examining the type-material nor sequencing the worms is "required" to transfer the species to the new genus, although these may be desirable.

- Thank you for your input regarding the terminology used and the process of transferring species to a new genus. We appreciate your suggestions. Regarding the term "status," your suggestion of using "classification" indeed holds merit, and we'll consider incorporating this change for clarity in our future revisions. As for examining the type-material and sequencing the worms for transferring species to the new genus, while it may not be an absolute requirement, our preference aligns with examining the type-material. Doing so helps ensure a comprehensive understanding and minimizes the potential for future synonyms. It enables a more accurate and robust transfer of species, contributing to the stability and clarity of taxonomic classifications.

333: "remarkably increased" and "increased remarkably" are not equivalent, I suspect the authors mean the latter here?

-Ok. It was corrected.

334: "based employed"?? "and barely,"??

 -Ok. It was corrected.

Reviewer 3 Report

Comments and Suggestions for Authors

This article makes a significant contribution to the understanding of the diversity of monogeneans. Among these parasites there are many species that are dangerous for fish, especially in artificial breeding. The authors of the manuscript justified the necessity of a new genus for two new species on the basis of molecular and morphological data. Species are described quite comprehensively.

However, I have some remarks on the manuscript, after correction of which the article can be accepted for publication.

 My comments are as follows

 1.       Title

“Proposal of Brotulella n. gen. from the gills of the Pacific bearded brotula Brotula clarkae Hubbs, 1944 (Ophidiiformes: Ophidiidae) based on morphological and molecular evidence”

Phrase genus from gills” has a vague meaning. It could be better phrased like this: Proposal of Brotulella n. gen. for monogeneans from the gills...

 2.       Introduction

The rationale for the purpose of the study, its relevance, and the importance of studying monogeneans should be expanded. Especially taking into account that the journal is dedicated to fish. Also, in the introduction the novelty of the study is not sufficiently substantiated, perhaps for this genus of fish or for this region. There is no information on whether monogeneans have previously been found in these fishes in this region and in general, or what is known about monogeneans parasitising them in other areas of the world ocean, which could indicate that they are poorly studied, etc.

This introduction is too brief.

 3.       Results

3.1. In Table 1 for the reference MG194743 the region is incorrectly indicated as Adriatic Sea, but in Table 3 from Dmitrieva et al., 2018 “Xenoligophoroides cobitis (Ergens, 1963) n. g., n. comb. (Monogenea: Ancyrocephalidae)..” https://doi.org/10.1007/s11230-018-9805-1 indicates: X. cobitis (BLACK8) - G. cobitis - Black Sea, off the Caucasus, Russia - MG194743.

Thus, this is not the Adriatic Sea, but the Black Sea.

3.2. “Brotulella n. g. resembles Ligophorus Euzet & Suriano, 1977, …. However, both genera differ from each other by having anchors with a stocking-shaped sheath associated with the distal end of superficial and deep roots (absent in Ligophorus spp.), two prostatic reservoirs with thick muscular walls (one pyriform prostatic reservoir without thick muscular wall in Ligophorus spp.) and by having a ventral bar bowed (V-shaped ventral bar in Ligophorus spp.).”

In description of both species and genus, it is not information about position of vas deferens relative to intestinal caecum, namely it is looping left intestinal caecum or not. This is an important diagnostic character, distinguishing some genera, including Ligophorus, from other genera of Ancyrocephalidae.  According to the figures, new genus, apparently, has vas deferens looping the intestinal caecum, that distinguishes it from Ligophorus.

Add this information both in description of genus and species, and in remarks to them.

Author Response

Response to Reviewer 3 Comments

This article makes a significant contribution to the understanding of the diversity of monogeneans. Among these parasites there are many species that are dangerous for fish, especially in artificial breeding. The authors of the manuscript justified the necessity of a new genus for two new species on the basis of molecular and morphological data. Species are described quite comprehensively.

However, I have some remarks on the manuscript, after correction of which the article can be accepted for publication.

 My comments are as follows

  1. Title

“Proposal of Brotulella n. gen. from the gills of the Pacific bearded brotula Brotula clarkae Hubbs, 1944 (Ophidiiformes: Ophidiidae) based on morphological and molecular evidence”

Phrase “genus from gills” has a vague meaning. It could be better phrased like this: Proposal of Brotulella n. gen. for monogeneans from the gills...

-Ok. It was made.

  1. Introduction

The rationale for the purpose of the study, its relevance, and the importance of studying monogeneans should be expanded. Especially taking into account that the journal is dedicated to fish. Also, in the introduction the novelty of the study is not sufficiently substantiated, perhaps for this genus of fish or for this region. There is no information on whether monogeneans have previously been found in these fishes in this region and in general, or what is known about monogeneans parasitising them in other areas of the world ocean, which could indicate that they are poorly studied, etc.

This introduction is too brief.

-Ok. It was modified.

  1. Results

3.1. In Table 1 for the reference MG194743 the region is incorrectly indicated as Adriatic Sea, but in Table 3 from Dmitrieva et al., 2018 “Xenoligophoroides cobitis (Ergens, 1963) n. g., n. comb. (Monogenea: Ancyrocephalidae)..” https://doi.org/10.1007/s11230-018-9805-1 indicates: X. cobitis (BLACK8) - G. cobitis - Black Sea, off the Caucasus, Russia - MG194743.

Thus, this is not the Adriatic Sea, but the Black Sea.

-Ok. It was modified.

3.2. “Brotulella n. g. resembles Ligophorus Euzet & Suriano, 1977, …. However, both genera differ from each other by having anchors with a stocking-shaped sheath associated with the distal end of superficial and deep roots (absent in Ligophorus spp.), two prostatic reservoirs with thick muscular walls (one pyriform prostatic reservoir without thick muscular wall in Ligophorus spp.) and by having a ventral bar bowed (V-shaped ventral bar in Ligophorus spp.).”

In description of both species and genus, it is not information about position of vas deferens relative to intestinal caecum, namely it is looping left intestinal caecum or not. This is an important diagnostic character, distinguishing some genera, including Ligophorus, from other genera of Ancyrocephalidae.  According to the figures, new genus, apparently, has vas deferens looping the intestinal caecum, that distinguishes it from Ligophorus.

Add this information both in description of genus and species, and in remarks to them.

-Ok. It was made.